# A Personalised Dietary Approach—A Way Forward to Manage Nutrient Deficiency, Effects of the Western Diet, and Food Intolerances in Inflammatory Bowel Disease

**DOI:** 10.3390/nu11071532

**Published:** 2019-07-05

**Authors:** Bobbi B Laing, Anecita Gigi Lim, Lynnette R Ferguson

**Affiliations:** 1Faculty of Medical and Health Sciences, University of Auckland, Auckland 1023, New Zealand; 2Nutrition Society of New Zealand, Palmerston North 4444, New Zealand

**Keywords:** inflammatory bowel disease, Westernisation, genotypes, nutrient deficiency, food intolerance, FODMAPs, gluten, fructose, lactose, brassica, mushrooms

## Abstract

This review discusses the personalised dietary approach with respect to inflammatory bowel disease (IBD). It identifies gene–nutrient interactions associated with the nutritional deficiencies that people with IBD commonly experience, and the role of the Western diet in influencing these. It also discusses food intolerances and how particular genotypes can affect these. It is well established that with respect to food there is no “one size fits all” diet for those with IBD. Gene–nutrient interactions may help explain this variability in response to food that is associated with IBD. Nutrigenomic research, which examines the effects of food and its constituents on gene expression, shows that—like a number of pharmaceutical products—food can have beneficial effects or have adverse (side) effects depending on a person’s genotype. Pharmacogenetic research is identifying gene variants with adverse reactions to drugs, and this is modifying clinical practice and allowing individualised treatment. Nutrigenomic research could enable individualised treatment in persons with IBD and enable more accurate tailoring of food intake, to avoid exacerbating malnutrition and to counter some of the adverse effects of the Western diet. It may also help to establish the dietary pattern that is most protective against IBD.

## 1. Introduction

The personalised dietary approach in combination with the knowledge of the genotype and genetic variants of an individual who has nutrient deficiencies, follows the Western diet, or may have food intolerances may be helpful for people who have inflammatory bowel disease (IBD). Gene–nutrient interactions could help explain the variability in response to food that is associated with IBD, and using this knowledge with a personalised dietary approach may offer a way forward. Identification of individuals’ genotypes is a first step to building up data sets which combine this information with data on individuals and their gut microbiome, dietary patterns, exercise routines, blood parameters and anthropometric measurements. This information can be used to build machine-learning algorithms to predict what foods/meals reduce inflammation and abdominal symptoms and maintain a healthy gut flora. Zeevi et al. have pioneered this approach in their work with diabetes [1]. They used this combination of information to successfully predict which foods/meals lowered postprandial blood glucose. Identifying genotypes associated with nutrient deficiencies or low nutrient absorption, the negative effects of the Western diet and the tolerance or intolerance of particular foods may also advise people on how to eat a greater variety of food. A wider range of food available to the gut microbiota enables their greater diversity, and this is associated with a healthy gut [2].

Pharmacogenetic research (the study of the variability in drug response due to genotype) is identifying gene variants with adverse reactions to drugs, and this is modifying clinical practice and allowing individualised treatment. The aim is to provide the right drug, at the right dose, at the right time, to the right patient [3]. Individual responses can be dependent on an individual’s genotype. Genetic variants can alter the metabolism of drugs as well as responsiveness, and this may necessitate changes to standardised drug regimes. The standard dose may not maintain the required therapeutic level if individuals metabolise the drug more quickly; conversely, if they are slow metabolisers the standard dose maybe toxic. Knowledge of pharmacogenetics and its application to precision medicine to enhance treatment responses and minimize drug-related adverse events is one approach that has been suggested as the way forward for IBD [4]. Pharmacogenetics information facilitates the identification of responders and non-responders to drugs and guides the decision about optimal treatment [5]. Similarly, further nutrigenomic research could identify nutrient–gene interactions in people with IBD. This would enable the more accurate tailoring of food intake and could suggest dietary patterns which avoid exacerbating malnutrition, curtail the adverse effects of the Western diet and explain intolerance to particular foods. This could lead to the minimization of abdominal symptoms and disease activity in those who experience IBD.

In an extensive study of 366,351 European individuals in 2016 aiming to identify dietary patterns and risk of IBD, no dietary pattern was identified with the risk of either of the two main expressions of IBD (i.e., ulcerative colitis (UC) or Crohn’s disease (CD)) [6]. However, when cases occurring within the first two years after dietary assessment were excluded, there was a positive association between a “high sugar and soft drinks” pattern and UC risk if they also had a low vegetable intake [6]. This study shows there are many challenges in determining the best dietary pattern for individuals with a particular chronic disease like IBD, given the genetic differences between people and their environments [7]. The current consensus is that no single nutrition recommendation can be made to modify this situation with respect to IBD—that is, there is no “one size fits all” solution for people with IBD [8]. However, using the personalised dietary approach combined with information on people’s genotypes may offer a way to resolve this conundrum.

The aim of this narrative review (based on searching two data bases, PubMed and Google Scholar) is to discuss the personalised dietary approach with respect to IBD. First it will identify nutritional deficiencies which individuals with IBD commonly experience, and give an example of a key nutrient whose absorption is affected by genotype. Secondly it will review the role of the Western diet in contributing to nutrient deficiencies and IBD, and give examples of some of the genes involved. Thirdly it will review the various food intolerances and food avoidances associated with IBD which may also contribute to nutrient deficiencies, and how particular genotypes can contribute to these.

## 2. Nutrient Deficiencies Associated with IBD

### 2.1. Nutritional Status of People with Inflammatory Bowel Disease

Three studies which have assessed the nutritional status of people with IBD are illustrated in Table 1. Assessment of people with IBD often shows they are under weight, with specific vitamin and mineral imbalances, and this contributes to their ill health [9,10,11,12,13,14]. The three studies in this table illustrate the common nutritional findings. Vagianos and colleagues collected dietary information by questionnaires (including dietary supplements) and observed multiple and clinically significant vitamin and mineral deficiencies in adults with CD who attended a gastroenterology outpatient clinic [9]. Inadequate reported intake was defined as <66% of the dietary reference intake (RFI) for each micronutrient. These authors reported that when comparisons were made by disease type or disease activity, there were no significant differences in the percentage of subjects consuming inadequate amounts of micronutrients. Stein and Bott (2008) reported on nutrient deficiencies in hospitalised adults with CD [10]. Hartman et al. in 2016 [15] investigated the nutritional status of children and adolescents who were ambulatory outpatients at a gastroenterology clinic. They reported on the percentage median nutrient intake compared with the required daily allowance (RDA) and dietary intake of healthy children. The intake was considered low if it was less than 80% of the RDA. During this study, two thirds of participants had active disease. However, there was no significant difference in nutrient intake between those with active disease and those in remission, and no association of nutrient levels with disease activity.

All three studies showed low levels of vitamin A and D and folate in people with CD. Stein and Bott [10] and Hartman et al. [15] both found substantial risk for deficiencies of magnesium, zinc and iron (Table 1). Stein and Bott [10] also showed lower B_12_ concentrations for inpatients and Vagianos et al. commented that in those with CD (compared to those with UC, the other main expression of IBD) had lower B_12_ concentrations with their serum biomarkers. Also, among men with CD there was a significantly higher proportion who had haemoglobin below the normal range compared to men with UC. Hartman et al. reported that anaemia was the most common sign of nutrient deficiency in their adolescents (51%). These studies illustrate that people with different expressions of IBD (i.e., CD or UC) have different nutrient needs. It also illustrates that it is important to assess nutritional status (through laboratory measurement of nutrient biomarkers) to avoid exacerbating malnutrition if particular foods are removed from the diet.

For people with CD in particular, malnutrition may also occur because of the severity of oral lesions experienced, which results in difficulties with tasting, chewing and swallowing. This means that some foods are avoided (e.g., tomatoes because of their perceived acidity). Oral involvement in CD has been identified in 8%–29% of people, irrespective of disease activity [16]. Examples of oral symptoms include labial swelling (which also may include gingiva and mucosa), mucosal polyps, cracking at the edges of the lips (angular cheilitis), ulcers, fissures and granulomas [17,18]. The mucous membranes in the mouth normally have a rapid cell turnover of 3–7 days, in contrast to the skin which is up to 28 days [19]. Loss of cells without replacement can lead to the mouth showing early signs of systemic disease and early predisposition to nutritional deficiency. This is another reason why optimal nutrition is important for people with IBD.

Many vitamins and minerals are considered necessary for healthy mucous membranes (Table 2) [20]. A number of these essential nutrients are low in people with people with IBD (Table 1 [9,10,15]). These nutrient deficiencies, and a higher presence of bacteria associated with dental caries, undoubtedly contribute to the oral symptoms experienced by people with CD. Sufficient levels of these nutrients would need to be included in any food regime suggested for IBD.

### 2.2. Nutrient Absorption Impacted by Genotype

One example of a nutrient whose absorption can be impacted by genotype is beta-carotene. The gene that affects its absorption is the beta-carotene 15,15′-monooxygenase gene (*BCMO1*). This gene encodes a protein which is an important enzyme in the metabolism of the nutrient beta-carotene to vitamin A [21]. It is expressed in a number of tissues (e.g., liver, lungs, skin, small intestine). Within the enterocytes of the mucous membranes, the enzyme cleaves beta-carotene into two retinal molecules. These are packaged with other dietary lipids into chylomicrons and then secreted into the lymphatic system. Fat in the diet enables optimal chylomicron formation [21,22]. According to the study by Leung et al., carrying one of the two polymorphisms (R267S: rs12934922 or A379V: rs7501331) of the *BCMO1* gene impedes the conversion of beta-carotene to retinol [23]. This study was based on a healthy population from the UK. Up to 45% of people in this study carried one of these two polymorphisms. Having one of the two polymorphisms of the *BCMO1* gene in conjunction with IBD could thus have an impact on the health status of the carrier. New Zealand (NZ) is a country with a high incidence of IBD (26/100,000 in 2016) [24], and in their most recent National Nutrition Survey (2008/9) reported that the NZ population sourced 58% of their vitamin A intake from carotenoids. The NZ population as a whole was also recorded as 22.7% and 12.1% deficient in vitamin A for males and females, respectively. For those aged 15-18 years, these percentages increased to 27.4% and 37.5% [25].

Adequate vitamin A intake is necessary for optimal activity of the innate and adaptive immune systems. Vitamin A is needed for the appropriate innate immune defence response. It affects neutrophil reactions to microbes, macrophage phagocytosis, and natural killer cell activity. Cultured human macrophages enriched with retinoic acid have been shown to have increased resistance to *M. tuberculosis* by down-regulating tryptophan–aspartate-containing coat protein (TACO) [26]. Vitamin A is also needed for the optimal formation of T and B cells in the adaptive immune system and the balanced type 1 and type 2 cytokine responses. Impairment of these functions through vitamin A deficiency leads to epithelial barrier dysfunction and inflammation—hallmarks of IBD [27]. Vitamin A deficiency in children increases the risk of signs and symptoms often associated with CD (i.e., impairment of iron use that aggravates anaemia, respiratory infections and diminished growth rates) [28]. If individuals with IBD also carry one of these at-risk alleles, they are at risk of being vitamin A deficient.

This may be a contributing factor to this nutrient deficiency reported in people with IBD. Using the personalised dietary approach and identifying if a person with IBD carries these variants linked to poor absorption of vitamin A would enable the tailoring of nutrient advice to ameliorate this. The Western diet may also contribute to nutrient deficiencies.

## 3. The Western Diet

The incidence of gastro-intestinal disorders like CD and UC has been increasing [29,30,31,32]. This has often been related to the current Western diet, which has changed substantially in the past seven decades [33,34,35,36,37,38,39]. This is because of changes in food production and technology, which provides easy access of urban based populations to cheap ultra-processed foods and foods high in refined grains, oil and sugar [40]. In Bangladesh, for example, the incidence of diagnosed IBD is very low. However, when Bangladeshis move to the United Kingdom where they are more highly exposed to Western culture and food choices, the incidence of IBD increases substantially within a generation [41]. This may also be associated with autoimmune disease developing as a result of increasing latitude, decreased exposure to sunlight and lower vitamin D intake [42]. They may also receive better health care which identifies their disease pathology more accurately. However, modern Western food is also becoming more available in the developing world. In the review of the global epidemiology of IBD by M’Koma, it was noted that there has been an increase in the incidence and prevalence of IBD in parts of Asia and Africa—particularly in urban areas and in higher socio-economic classes [43]. This Westernisation of the diet has resulted in lower intakes of dietary fibre, increased consumption of saturated fatty acids, and sugar. Westernisation has also introduced other factors such as changes in birthing methods, increased use of artificial formula instead of breast feeding, and the use of sugar substitutes, food additives and antibiotics. Many of these changes have been associated with a reduction in gut microbial diversity, which is a feature of IBD [44,45].

### 3.1. Lower Intakes of Dietary Fibre

As the diet becomes more Westernised and processed, “Western” food often contains less fibre. Current intakes of fibre associated with the Western diet are assessed to be about 20 g/day, compared to a fibre content of 70–120 g/day when diets rich in fruit, vegetables and grains are consumed, as is estimated for hunter-gathers and agriculturalists in human history [46]. The low consumption of fruit, vegetables and dietary fibre and the increased intake of animal fat and refined sugars have been observed in people with disorders like IBD [47], as are exclusion diets for IBD that are made popular in the media [48].

The risk of IBD has been negatively associated with the intake of fibre [34,49,50]. Fibre can be defined in a number of ways, as illustrated by the different definitions by the American Association of Cereal Chemists (AACC) [51] and The Codex Alimentarius Commission [52]. A recent AACC definition describes fibre as being a component of “edible plants or analogous carbohydrates that are resistant to digestion and absorption in the human small intestine, with complete or partial fermentation in the large intestine” [53]. They include in their definition polysaccharides, oligosaccharides, lignin and associated plant substances. Fibre has been further subdivided into rapidly digested starch, slowly digested starch and resistant starch (RS), and the latter is classified into five categories (RS1–RS5). High levels of resistant starch are found in unripe bananas, raw potatoes, maize, pulse grains and legumes. However, depending on their processing, the RS content can vary, and in legumes this can range from a few percent to 80% [54]. The RS3 category (retrograded starch) has been shown in wild type interleukin-10-deficient IBD murine studies to decrease ileal and colonic inflammatory lesions compared to a low-fat diet [55,56]. The authors suggested that soluble fibre and resistant starch suppress gut inflammation. The inclusion of soluble fibre and resistant starch in the diet of individuals with IBD may be helpful for this reason.

Fibre can also be described as soluble and insoluble [57]. The removal of insoluble fibre, a component of the skins of fruit, vegetables, whole grains, and seeds, is a current recommendation by the Crohn’s and Colitis Foundation of America [58]. When people are experiencing disease activity (sometimes described as a flare or relapse), insoluble fibre may intensify bloating, pain, diarrhoea and gas. However, the consumption of soluble fibre in vegetables (e.g., the brassica family, dried peas and beans, parsnips, carrots, potatoes, spinach, squash, zucchini) and fruit (e.g., pip and stone fruit, berries, bananas, dates, cantaloupe, grapes and pineapple), oat cereals and barley is recommended. Soluble fibre promotes water absorption, enabling the gut contents to form a gel-like consistency, which delays emptying of the gut and reduces diarrhoea [58]. Soluble fibre also provides a substrate for bacterial fermentation in the colon [59]. The soluble plant fibres in plantain and broccoli reduce the movement of *E. coli* across M cells in Peyer’s patches in CD patients [60]. In CD, *E. coli* is found in increased numbers in the mucosa [61,62]. This means it is important that appropriate fibre types are chosen if people with IBD are removing food or food groups (e.g., selecting gluten-free diets or foods low in FODMAPs (fermentable oligo-, di- and mono-saccharides and polyols)) so they do not inadvertently contribute to an increase in disease activity later in their health journey [63].

There are also identified gene–nutrient interactions associated with the role of fibre. One of these is the free fatty acid receptor 2 gene (*FFAR2*; previously known as G protein-coupled receptor 43—*GPR43*). This gene is a member of the largest gene family—the seven-transmembrane receptor group known as G-protein-coupled receptors (*GPCRs*) [64]. These are involved as mediators of cellular responses to many neurotransmitters and hormones [65]. *FFAR2* is highly expressed in neutrophils [66]. One pathological marker of CD is the migration of neutrophilic granulocytes into the mucosa of the intestine [67]. *FFAR2* also activates pathways involved in intracellular Ca^2+^ release and inhibits the build-up of cAMP [68].

The *FFAR2* receptors are present in fat tissues, inflammatory cells and the large intestine [69]. In the large intestine, the receptors are stimulated by short-chain fatty acids (SCFAs), which are produced by bacteria fermenting dietary fibre [21,70]. Lower intakes of dietary fibre decrease the production of SCFAs, which stimulates *FFAR2* signalling. The study by Sivaprakasam et al. showed the critical role of *FFAR2* in the maintenance of healthy gut microbiota, leading to the suppression of intestinal carcinogenesis [71].

The interaction of SCFAs and FFAR2 greatly affects inflammatory processes. SCFAs play an important role in colon homeostasis, as they promote pathways involved in cell differentiation, growth regulation, immune regulation and apoptosis [72]. SCFAs and butyrate in particular seem to be the favoured fuel source of the bacteria in the colon as well as of colonocytes, which are required for the functional integrity of the epithelium [73,74,75,76]. The source of SCFAs is the intake of highly fermentable fibres, the residues of which are converted by anaerobic bacteria into SCFAs upon reaching the large bowel [77]. SCFAs have been observed to increase the beneficial populations of *Bifidobacterium* and *Lactobacillus*, which have a role in anti-inflammatory outcomes and immune functions [38,78,79]. SCFAs appear to be important in epithelial barrier maintenance and immune system regulation. Maslowski and Mackay consider the SCFAs produced in the colon to be very important for immunoregulation. They propose that the lower dietary fibre intake associated with the modern Western diet is the driver behind the increased incidence of inflammatory diseases associated with this lifestyle factor [70].

However, there is currently no published literature about the significance of this gene with respect to people with IBD and their intake of fibre. People with IBD (whether with disease activity or in remission) have been observed to have a low intake of fruit, vegetables and dietary fibre [49,80]. Perhaps particular variants in the *FFAR2* gene may affect a person’s response to fibre, and this in turn contributes to changes in microbiota composition. This may explain how those who follow a strict FODMAP diet experience a reduction in beneficial bacteria [81,82].

### 3.2. Increased Consumption of Saturated Fatty Acids

Some authors believe that the Western diet has also become pro-inflammatory as a result of the increased consumption of saturated fatty acids and decreased consumption of polyunsaturated fatty acids (PUFAs), especially long-chain PUFAs (LC-PUFAs) [83,84,85,86,87]. There has also been a decrease in the ratio of the omega-3 and omega-6 PUFAs ingested, partly because of a lower consumption of fish. Fish intolerance and allergy is well-recognised in childhood, being the third most frequent allergen after cow’s milk and egg in European children [88,89,90]. There is less evidence implicating fish intolerance or allergy in CD. Untersmayr et al. observed that antacid medication plus fish consumption impeded the digestion of dietary proteins in their BALB/c mouse fish allergy model [91].

In earlier times, people consumed only small amounts of these PUFAs, and in about the same proportion. Today it is estimated that this ratio is now about 1:10–1:20–25 omega-3:omega-6. This has occurred from the increased intake of vegetable oils (e.g., soy, safflower, corn and sunflower) [83,92,93]. The genes and their variants linked to differences in fatty acid absorption include fatty acid desaturase genes (*FADS1*, *FADS2),* the peroxisome proliferator-activated receptor genes (*PPARA, PPARG),* the X-ray repair cross-complementing protein 1 gene **(***XRCC1*) and stearoyl-CoA desaturase (*SCD1).* A number of studies have discussed the role that these genes play in the serum levels of LC-PUFA-omega-3 and omega-6 fatty acids, as well as the effects of various variants on metabolic pathways and how this impacts inflammation and cancer risk [94,95,96,97,98,99,100,101,102,103,104]. These studies show that the impact of these fatty acids—especially on inflammation—is also influenced by the person’s genotype. This may have deleterious effects on individuals with IBD who exclude fish and fish oils from their diet and have a high omega-3:omega-6 PUFA ratio.

### 3.3. Fructose and Sugar Consumption

There is much debate in the literature about the effects of fructose consumption [105,106,107,108,109,110,111,112,113,114,115,116]. The rise in obesity, diabetes, metabolic syndrome, IBD, coronary heart disease, and whether the increased consumption of fructose or fructans has contributed to these disorders, has been widely discussed [105,106,107,108,109,110,111,112,113,114,115,116]. Over recent decades, increasing amounts of fructose have been consumed as part of the Western diet. This has been particularly associated with the addition of sugars to processed foods, particularly high-fructose corn syrup (HFCS, comprising ~60% fructose) in the USA or with cane sugar (50% fructose) in NZ. Both are cheap and ubiquitous sweeteners, especially in soft drinks and cereal products [117]. The effect of fructose in infant formulas on vascular inflammation has also been investigated in full-term babies. The study found that the fructose formula had a higher inflammatory response than the fructose-free formula, but the difference was not significant [118]. The authors commented that the result may become significant in a preterm infant [118]. According to recent USDA figures, the estimated calories of HFCS consumed daily has risen from 0.4 g/day in 1970, peaked at 44.3 g/day in 1997, and has gradually declined (31.0 g/day in 2015) [25,119,120]. However, total “sweetener” consumption has increased compared to the 1970 figures (from 87.3 g/day in 1970 to 94.1 g/day in 2015) [119]. In NZ, the Adult Nutrition surveys of 1997 and 2008–2009 report that the mean fructose intake and the mean total sugar consumption for European males and females has declined from 1997 (26 g/day and 139 g/day, respectively), to 2008 (23.4 g/day and 100.3 g/day respectively) [121]. Unfortunately, there are no recent national consumption figures to see whether that national trend has continued. A small regional study (*n* = 227) in 2014 on a Canterbury (NZ) population aged 50 years reported that the mean intake for fructose for men was 25.5 g and 20.6 g for females [122]. This suggests that the downward trend has not continued.

Once ingested, fructose is transported into the blood stream through sugar-transporting proteins GLUT2 and GLUT5, via the small intestine in limited amounts to the liver. The liver removes some of the fructose from circulation to maintain the correct proportion of glucose to fructose (10 times lower) [123,124,125,126], and converts fructose into other metabolites including triglycerides and uric acid. This latter function is thought to be a contributor to the rise in obesity, metabolic syndrome and diabetes [106,110,127,128]. Fructose intake also stimulates the thioredoxin-interacting protein gene (*TXNIP*), which can lead to higher fructose intake [129]. The *TXNIP* gene has been associated with promoting inflammation in endothelial cells, mediating hepatic inflammation, and regulating NF-κB [130,131,132,133,134]. *TXNIP* has been shown to be involved in the differentiation of epithelial cells in the intestine [135]. *TXNIP* mRNA, which negatively regulates the *TXNIP* gene, shows decreased expression in the inflamed mucosa of the colon in people with UC [136]. Interestingly, thioredoxin-1 produced by the gene *TXNIP* is also inhibited by the production of vitamin D3 up-regulated protein 1 (VDUP1) when vitamin D_3_ is given [137,138].

The relationship between fructose uptake and the *TXNIP* gene may explain why fructose intake has been increasingly linked to intestinal symptoms of bloating, abdominal pain, diarrhoea and fructose intolerance in individuals with IBD [126,139,140,141,142]. This underlies the need to consider fructose intake when managing these symptoms. A recent NZ study by Spencer et al. on people with irritable bowel syndrome (IBS) (*n* = 277) based in Canterbury, NZ found that the participants’ IBS score, constipation and diarrhoea were not associated with fructose intake. However, they found that greater mean daily fructose intakes (*p* = 0.05) were associated with a lower prevalence of IBS pain [122].

In adults, fructose intolerance is most commonly assessed by breath testing after swallowing 25 g of fructose dissolved in water. Breath measurements are made on a 30-min schedule for up to three hours. A value of ≥20 ppm H_2_, accompanied by symptoms (e.g., bloating, diarrhoea), is an affirmation of fructose intolerance [139,143]. This is because the hydrogen produced by gut bacteria is reacting with the fructose which has not been absorbed in the small intestine. Breath testing is a common method of testing for fructose, and there are a variety of methodologies in its application, as well as questions about its reliability [144]. A North American consensus statement has been drawn up for hydrogen and methane-based breath testing in gastrointestinal disorders to address these issues [145].

When it is suggested that high-fructose-containing foods be eliminated in the diet, care must be taken in deciding which foods are excluded, as the assessment of what constitutes a high-fructose food varies [146]. If people stay on a low-fructose diet for too long, they may be in danger of nutrient deficiencies due to lack of fruit and vegetables in the diet. It also needs to be remembered that fructose taken as a bolus (e.g., after swallowing 25 g of fructose in a fructose intolerance test) is metabolized differently to when it is ingested in the presence of glucose. Many people absorb fructose easily along with sucrose or with glucose that is at similar levels [147].

### 3.4. Other Western Diet Factors

Westernisation has also introduced other factors into the diet, such as changes in birthing methods, using artificial infant formula instead of breast feeding, and the use of sugar substitutes, food additives and antibiotics (Table 3). Many of these changes have been associated with a reduction in gut microbial diversity as well as dysbiosis (i.e., microbial imbalances in the gut)—a feature of IBD [44,45]. This path to dysbiosis may start early, with lack of exposure to appropriate gut microbiota as an infant. The composition of an infant’s gut bacteria is affected by the delivery method (i.e., whether infants are born by caesarean (C-section) or vaginally, and whether breast-fed or given infant formula. C-section babies have microbiota that resembles that on the mother’s skin, and those born vaginally are seeded by the microbiota in the mother’s vagina [148,149]. The microbiota seeded from the mother’s vagina is thought to contribute to a normal infant microbiome, whereas having a C-section and all the factors associated with the reason for having a C-section does not [150].

The type of first foods also contributes to the composition of the microbiota. Breast feeding promotes immune development and protects against many diseases. In a small study on 3-month-old babies 1214 probe sets for epithelial cells were significantly differently expressed in exclusively breast-fed (*n* = 12) and formula-fed babies (*n* = 10) [151]. This may explain some of the differences that have been noted in clinical and epidemiological observations of these two groups. The two groups have different gut development, with the breast-fed babies being described as being “less leaky”. Bacteroidetes species, which are important players in the commensal bacteria, were not found in the stools of the formula-fed infants [151]. Breast milk also contains human milk oligosaccharides (HMOs) which stimulate the growth of the intestinal flora, acting as prebiotics for beneficial bacteria [152]. Human bioactive milk proteins modulate the immune system, and this too would contribute to the enhancement of gut barrier function [153,154]. Formula feeding is becoming an increasingly common pattern of infant feeding in Western culture.

The increasing use of artificial sweeteners (ASs) may also contribute to dysbiosis [155]. A number of studies have been conducted on the effects of ASs on the gut. These have looked at their effects on secretion, absorption, gut motility, the microbiome and gastrointestinal symptoms [156,157,158,159,160]. Some studies imply that ASs have measurable effects on the oral/gut microbiota [155,161,162]. In one example, male rats were randomised into control and AS groups and were exposed to doses ranging from 1.1 to 11 mg/kg/day for 12 weeks (the USA Federal Drug Administration (FDA) acceptable daily intake for humans is 5 mg/kg/day). At the low dose of 1.1 mg/kg, the researchers reported that total anaerobes were reduced by approximately 50% in all sucralose plus maltodextrin groups. However, when this study was reviewed by an expert panel [163], it was criticised for not having an isocaloric carbohydrate control, and not correcting for the water content of the stools. Both these factors can affect the interpretation of the bacterial concentrations. This report also commented that the difference between control and treatment groups for bacterial counts given were also less than 10-fold. They concluded that this meant that the results could not be interpreted as indicative of adverse change, but reflected the normal range of variation in bacteria over this period of time. However, further research in healthy human subjects continues to suggest that microbiota are adversely affected by ASs. Suez and colleagues showed that with 11 weeks of exposure to three common ASs, saccharin altered gut microbiota and induced glucose intolerance [161]. This indicates the need for more studies—particularly in humans—to elucidate the effects of ASs on microbiota [164,165,166].

Two other recent additions to the Western diet are emulsifiers in food to improve mouth feel in low-fat products, and antibiotics in animal feed [167,168]. Emulsifiers have been shown in murine studies to have an impact on the mouse gut microbiota that promotes colitis and decrease in (alpha) diversity [167]. Antibiotics are used to reduce disease in livestock kept in confined quarters. Repeated courses of antibiotics can lead to the development of what is termed “the resistome”—the collection of genes in bacteria that confers resistance to antibiotics [169]. Increased antibiotic exposure in humans through food sourced from animals that have had antibiotics may affect the human biome. When these products from these animals are consumed, they contribute to antibiotic resistance. The US Food and Drug Administration (2009) indicated that 80% of antimicrobial products in the USA were used in animal feed as growth stimulants, not for veterinary use [170]. In the highly pastoral country New Zealand it is reported that about 60% of antibiotics are administered to animals [171]. The Ministry for Primary Industries (NZ) states the use of antibiotics in cattle and sheep is low compared to the intensive rearing pig and poultry industries [172].

The issue for people with IBD is its effect on the genetics of their microbiome when they incorporate these products into their diet. For example, with antibiotics, antibiotic resistance genes are built up in the microbial community both in pathogenic and non-pathogenic bacteria, and this builds the resistome [169]. Genes in the resistome are readily transferred not only from one bacterial cell to another, but also by cassettes of multiple genes [173,174]. This has been shown to result in a build-up of multidrug resistance to *E. coli* in domestic animals [175,176]. These bacteria have also been identified as being in increased numbers in those with CD [177,178,179]. Collectively, all these Western diet factors impinge on the genetics of the biome and reduce microbiota diversity. Personalising the dietary approach for people with IBD and identifying the effects of genes associated with these changes (Table 3) would help reduce this adverse impact on the genes of the microbiome. It would help maintain microbiotal diversity and positively modulate disease activity.

## 4. Food Intolerances

IBD has been linked with a number of food intolerances, especially with foods containing gluten, dairy products/lactose or fructose (discussed earlier), or food high in FODMAPs. Many of these intolerances can also be linked to specific genes or genotypes. Other food intolerances linked with genotypes and IBD are brassica vegetables, mustard, wasabi, raw and cooked tomatoes, sweet potatoes, mushrooms, sulphur dioxide, sulphites and sulphur compounds. The specific genes or genotypes associated with these intolerances will be discussed further. This section begins with a discussion of the FODMAP approach, as this is the newest dietary strategy suggested as being helpful for IBD. It also links to a number of food intolerances associated with particular genotypes and/or gene variants.

### 4.1. FODMAPs

FODMAPs are fermentable oligo-, di- and mono-saccharides and polyols which, if lowered in the diet, have been shown to be effective in decreasing the abdominal symptoms associated with IBS. This approach is being proposed as being helpful also for people with IBD [2]. Foods that are identified as FODMAPs have been discussed extensively in the literature [111,180,181,182,183,184]. Table 4 gives examples of food sources of FODMAPs [185,186,187,188,189,190,191] and genes that are associated with them.

Foods high in FODMAPs have been characterized in a healthy population (i.e., those who do not experience inflammation) as not being well absorbed in the small intestine, having a laxative effect if taken in higher doses and fermented rapidly by bacteria [2]. Inflammation of the gut wall—a characteristic of IBD—also interrupts enzyme production by the brush border cells, and this also contributes to the sugars not being absorbed. Unabsorbed sugars go to the large intestine, where bacteria interact with them. When some metabolites (e.g., butan 2,3 diol, ketones, acids, aldehydes) are formed, they can result in an alteration of the signalling mechanisms of the bacteria and this causes gas, pain and diarrhoea [192]. Diarrhoea has a negative impact on water and electrolyte balance, and may lead to dehydration. Gas, pain and diarrhoea commonly occur in non-IBD gastro-intestinal tracts (i.e., healthy populations) if doses of oral fructose or polyols over 50 g are taken [113,193,194].

This physiological reaction is the rationale for using the FODMAP dietary regime in gastrointestinal disorders like IBD [111,195]. This is based on studies with irritable bowel syndrome (IBS), where the decreased consumption of these fermentable carbohydrates has decreased abdominal symptoms (i.e., pain, flatulence, bloating, constipation, diarrhoea). There is no question that the by-products of FODMAP digestion and absorption may be especially unpleasant, exacerbating the symptoms of individuals with gastrointestinal disorders. Thus, it might appear as self-evident that avoidance of this group of compounds could be therapeutic for individuals with IBD—especially if they also have IBS [111,195]. A number of trials with IBS have suggested that this approach shows potential for improving symptoms in IBD. However, to avoid compromising people’s nutritional status, care must be taken so that any changes instigated in sources of fruits and vegetables do not reduce the total recommended intake of fruits, vegetables or fibres [196].

How does the FODMAP approach relate to genotypes? As will be discussed further on, intolerance to some vegetables (e.g., brassica, mushrooms, cooked tomatoes) whole grains and dairy products may be linked to specific genotypes and gene variants (Table 5). Recent research has also indicated that following a strict FODMAP diet can also have an effect on the genome of the microbiome and encourage harmful gut microbiota and a decrease in *Bifidobacterium* [183,184]. In people with disorders like IBD, their microbiotal composition is different from those found in healthy people [197,198,199]. Individuals with IBD have reduced numbers of microbiota, fewer divergent species with lower numbers of *Firmicutes*, and increased number of *Proteobacteria* species compared to healthy people [200,201,202,203]. This means that strict adherence to a low-FODMAP diet could contribute to this dysbiosis.

In addition, people with IBD may have a loss of function of the fucosyltransferase 2 (*FUT2*) gene, which identifies them as a “non-secretor”. A non-secretor has a significant reduction in microbiotal diversity, richness and abundance [204]. The *FUT2* gene has been identified as increasing the susceptibility to CD [205,206,207]. Non-secretor individuals are also infrequently colonised by *Bifidobacterium bifidum* (important in the infant flora), *B. adolescentis* and *B. catenulatum/pseudocatenulatum* (important in adult intestinal flora) [204]. Bifidobacteria are an important part of the human intestinal microbiome. They are crucial to the development of a healthy infant microbiota, encouraging appropriate gut maturity and gut integrity, modulating immunity and negating the effects of pathogens [208,209]. This would suggest that people with IBD who are non-secretors have an increased risk of having a less-effective defence against a number of pathogens and are consequently more vulnerable to having an inflamed gut. Following a low-FODMAP diet may exacerbate this. Further research is needed on those who are a non-secretors, using the low-FODMAP approach to see how this affects their microbiota.

### 4.2. Gluten

Intolerance to foods containing gluten is associated with the autoimmune disease coeliac disease and with non-coeliac gluten sensitivity. These two have symptoms in common (e.g., bloating, diarrhoea, constipation, abdominal pain and weight loss). Unlike coeliac disease, gluten sensitivity is not associated with antibody formation and damage to the epithelium of the gut. Coeliac disease has genetic links. Gluten intolerance is associated with gene variants of the major histocompatibility complex, class II, DQ alpha 1 (*HLA-DQA1*) gene and the major histocompatibility complex, class II, DQ beta 1 (*HLA-DQB1*) gene [210]. Festen and colleagues in their meta-analysis of two genome-wide association studies GWAS also identified the genes interleukin 18 receptor accessory protein (*IL18RAP*); phosphatase, non-receptor type 2 (*PTPN2*); T cell activation RhoGTPase activating protein (*TAGAP*) and pseudouridine synthase 10 (*PUS10*) as having shared risk loci for CD and coeliac disease [211]. The gene *PTPN2* is involved in the regulation of the epithelial barrier function and is modulated by the vitamin D receptor gene (*VDR*) [212]. Current nutrition advice for people with IBD includes checking whether they have the autoimmune coeliac disease or if they have non-coeliac gluten sensitivity. The blood test to detect the presence of anti-tissue transglutaminase (tTG) antibodies can be taken. These antibodies are produced in the small intestine when inflammation occurs in response to gluten. Another test that has a high specificity for coeliac disease is the anti-endomysial antibody (EMA) test which detects villus atrophy—a feature of coeliac disease [213]. If IBD patients prove to have an adverse response to gluten from these tests, then avoiding these foods and products containing them is advised. These foods include wheat, oats, barley (which includes malt and malt vinegar) and rye. There is no definitive test for non-coeliac gluten sensitivity except for eliminating gluten from the diet to see if symptoms subside. The role of non-coeliac gluten or wheat sensitivity in the general population as well as in IBD has been debated extensively. A review of the topic suggests that it may be linked in IBD with those who also have IBS.

### 4.3. Dairy Products and Lactose

The sensitivity to dairy foods in people with IBD is about 10%–20% [214,215,216]. Some studies have also linked increased incidence of IBD with the increased intake of dairy foods [217], particularly cheese [218]. This may be because of the fat content of cheese [219]. Nolan-Clark and associates noted that in their study on CD, sensitivity to dairy products was not related to disease activity status [219]. In a year-long trial on UC conducted by Wright et al. it was noted that patients had fewer relapses on a milk-free diet [216].

This sensitivity to dairy products is often associated with the reduction of lactase enzyme in adulthood. Lactase is an enzyme located in the villus enterocytes that is required for the digestion of lactose in milk. Lactase catalyses lactose, the disaccharide in milk, to galactose and glucose (i.e., monosaccharides which can be absorbed into the blood stream). In most mammals, its activity declines shortly after weaning. In humans, the enzyme may persist with its activity into adulthood. This is known as lactase persistence, and it is a genetically determined trait. In Europe, a single allele—T-13910 of the lactase phlorizin hydrolase (*LCT*) gene—is the main one. However, in Africa and the Middle East many more mutations are associated with lactase persistence [220,221].

This lactase persistence has associations with CD. In an NZ-based study [222], lactase persistence in the Caucasian population was associated with an increased risk of CD for those with the lactase (*LCT*) gene who were homozygous for the T allele of rs4988235 as compared with those homozygous for the C allele (OR = 1.61, 95% CI: 1.03–2.51). Increasing lactase non-persistence in populations has also been linked to a decreased risk of CD [223]. In a study encompassing populations from 26 countries around the world, CD decreased as lactase non-persistence increased (*p* < 0.01), [223].

Not consuming dairy products raises the risk of having insufficient intakes of calcium in the diet, especially if few other sources of calcium are consumed. Calcium is not only essential for bone health, but Ca^2+^ is extensively used in biological messenger systems and cell functions [224]. Low calcium intake of people with Crohn’s disease increases their risk of bone demineralisation and osteoporosis. These are well-known complications of this condition. Bernstein reported for people with IBD that “The overall relative risk of fractures is 40% greater than that in the general population and increases with age” [225]. Osteoporosis can also be exacerbated by the use of steroid medications (a common treatment in IBD), as they have a detrimental effect on calcium and phosphate metabolism [226]. Calcium intake is important with respect to IBD because, as discussed earlier, Vagianos et al. observed that 25% of their adult outpatients consumed less than the recommended intake of calcium. Further, the study of Hartman et al. showed that 79% of the adolescent outpatients consumed less than 80% of RDA [9,15]. If people are not tolerant to dairy foods, non-dairy sources of calcium are available, such as fortified soy, almond and rice milks, tofu, sardines, almonds, sesame seeds, broccoli and fortified breakfast cereals and juices [227].

### 4.4. Brassica

Vegetables such as broccoli, cabbage, Brussels sprouts and cauliflower are members of the brassica family of vegetables. These are known to be a good source of various micronutrients (e.g., selenium, iodine) [228] and various phytochemicals (e.g., glucosinolates and quercetin) [229]. Common members from this group of plants are recommended to be excluded from the low-FODMAP diet (Brussels sprouts, broccoli and cabbage) [195]. However, two randomized controlled diet studies conducted by Brauer et al. found different responses in individuals depending on their glutathione S-transferase Mu 1 (*GSTMl*) genotype with supplements of specific brassica vegetables (broccoli, cabbage, cauliflower and radish sprouts) [230]. Their study found that this combination of brassicas altered the circulating peptides transthyretin (TTR) and zinc alpha 2-glycoprotein (ZAG) fragments in a sensitive and identifiable manner. TTR declines in serum with early signs of inflammation. ZAG (an adipokine) is mainly involved in lipid mobilisation. In *GSTMl+* individuals, ZAG and TTR declined with brassica intake. However, this did not occur in those with the *GSTMl-nul1* genotype.

*GSTMl* and genotypes of *GSTTl* are also significant with respect to individuals and their intakes of brassica. There have been over 500 studies on GST enzymes showing that genotypes of these *GSTMl* and *GSTTl* have an influence on cancer outcomes, contingent on the expression or non-expression of these genes [231]. *GSTT1* and *GSTMl* belong to the GST gene super family, which is an extensive group of enzymes. The *GST* family contains several classes (e.g., alpha, kappa, mu, omega, pi, sigma, theta and zeta) [232,233]. GST enzymes have significant roles in the metabolism of electrophilic compounds (e.g., drugs and carcinogens) [234].

At the *GSTMl* locus, four allelic variants have been discovered in humans, and the null allele is present in about 50% of individuals [235]. Where the *GSTMl* gene deletions or the null genotype exist, there is a total lack of the protein encoded by the gene [236]. People with deletions are thought to be more vulnerable to toxic xenobiotics (e.g., antibiotics and drugs) [237]. There is also a variation between different cultural groups in the frequency of the *GSTMl* deletion. In Caucasian and Asian populations, the frequency of the *GSTMl* null genotype genetic polymorphism is estimated to be about 53%. However, the frequency is lower in African- American populations, at 27% [231].

GST enzymes can also be acted on by isothiocyanates (found in brassicas) either as inducers or as substrates in liver detoxification pathways, so a deletion or lowered activity of a *GST* gene can result in a persistent or elevated level of isothiocyanates in the body [238]. Studies based in Asia suggest that the null or less-active genotypes for *GSTT1* and *GSTM1* are protective and lower the risk of colorectal cancer [239]. However, the opposite occurred in the USA-based studies. The active or expressed genotypes, not the null genotypes, were associated with higher risk reduction [240]. The disparity of these results may reflect the different glucosinolate content of the brassica consumed. In Asia, Chinese cabbage (bok choy) is the predominately eaten brassica, and its main glucosinolate is glucobrassicin, whereas in the USA where raw broccoli is the more favoured brassica, the glucosinolate contents differ [241]. The main glucosinolate in this vegetable is glucoraphanin [241]. Glucobrassicin is an indole, whereas glucoraphanin is a sulforaphane. Their distinct chemical structures as shown in Table 5 (adapted from Higdon et al.) may mean that they are metabolised differently as they pass through the intestine, and this would suggest that their influence on the GST genes could also be dissimilar.

A pilot study by Campbell et al. [242] was conducted on the *GSTT1* (-/-) deletion in people with CD, in a New Zealand cohort from the Auckland IBD study [243]. This also showed those consuming broccoli, cauliflower and Chinese greens were more likely to report experiencing a beneficial effect if they had the *GSTT1* deletion. This may well apply to people with other gastrointestinal disorders such as IBS [244]. Responses to another brassica, rocket (arugula), have also been found to be dependent on a variant in the single-nucleotide polymorphism (SNP) rs9469220 of the human leucocyte antigen gene (*HLA*) in people with CD [245]. An inflammatory colonic phenotype has also been associated with variants in the *HLA* gene region [246,247,248]. This polymorphism has been linked to total IgE levels, which have a key role in sensitivity type 1 reactions in allergies linked with rhinitis, dermatitis, asthma and urticaria [249,250,251]. This adverse reaction to rocket in people with CD suggests that consumption of this food initiates the immune response in a type I sensitivity reaction. Another variant in the SNP rs7515322 in the iodothyronine deiodinase 1 gene (*DIO1*) has also been associated with an adverse response to broccoli. *DIO1* is part of the selenoenzyme family, which are important as signalling molecules for thyroid hormones and pivotal to thyroid metabolism. Both selenium and iodine deficiencies exist in NZ, and this may heighten individuals’ adverse response to broccoli if they have this variant [245]. This influence of micronutrients may also explain inter-country variation in responses to broccoli.

### 4.5. Mustard, Wasabi, Raw and Cooked Tomatoes and Sweet Potatoes

In another study by Marlow et al. [252], tolerance of mustard, wasabi, raw and cooked tomatoes and sweet potatoes was dependent on the G variant of the SNP rs12212067 in the *FOX03* gene. Seventeen foods were identified with beneficial effects, and four foods with detrimental effects. Sweet potatoes had the highest reported frequency of beneficial effects, and mustard, wasabi, raw and cooked tomatoes were detrimental for over 25% of the study population with the G allele (SNP rs12212067) associated with the *FOX03* gene. FODMAP guidelines suggest that tomato products such as canned tomatoes are not appropriate in the low-FODMAP diet, although plum tomatoes are [139]. This guidance could perhaps be tailored more for people using the low-FODMAP diet if their gene variants in the *FOX03* gene were identified.

### 4.6. Mushrooms

In a study on CD in an NZ population, a statistically significant association with a variant (the T allele) of the SNP rs1050152 of the organic cation transporter gene *OCTN1* was found. This variant is related to mushroom intolerance in those with CD, when compared with the general population. This indicates that NZ people with CD who have this variant have an increased risk of adverse symptoms associated with the intake of mushrooms [244].

### 4.7. Sulphur Dioxide, Sulphites and Sulphur Compounds

Sulphur dioxide and sulphites (at concentrations greater than 10 mg/kg or 10 mg/L) are now listed by the EU as a common cause of food intolerances and allergens [253]. Additive sulphites are often used in the food industry, as they have multiple functions. They are used as food preservatives, as they stop microbial growth and reduce the spoilage of food. They can act as antioxidants, inhibiting browning actions. They can modify texture in doughs. They are found in wine, soft drinks, sausages and burgers [254]. There is some evidence that they may aggravate IBD symptoms. Magee et al. report in their novel method of diet analysis that in UC, sulphites adversely affect symptoms through their anti-thiamine activity. Thiamine has a critical role in energy metabolism and cell function [255]. Low levels have been implicated in fatigue and IBD in those in remission (UC) or quiescent (CD) [256]. Those with very active disease who ate large quantities of food containing sulphites also had high food sigmoidoscopy scores [257].

Earlier studies linking sulphur compounds with IBD have described hydrogen sulphide as a bacterial toxin in people with UC [258]. This description was based on the experimental work where sulphated polysaccharides were ingested by guinea pigs and rabbits [259,260]. These sulphur products impaired butyrate oxidation—a characteristic of UC [261]. Butyrate, as a SCFA (as mentioned earlier), plays an important part in colon homeostasis by promoting the development of the intestinal barrier through enhancing tight-junction formation, which is impaired in people with IBD [262]. More recent work by Carbonero et al. has summarised the significance of microbiota in colonic sulphur metabolism and identified a number of gut microbiota taxa that participate in it [263]. These authors suggest that common variants in host genes involved in the sulphide oxidation pathways may be implicated in ineffective epithelial sulphide detoxification. This makes people with these variants predisposed to sulphide intolerance and consequent inflammation. The literature suggests that sulphur dioxide sulphites, and other ingested sulphur-containing products, are another example of where nutrient–gene interaction occurs. Particular variants are associated with their intolerance. Using the personalised nutrition approach and identifying these gene variants in people with IBD would identify those who are intolerant to sulphur dioxide, sulphites and sulphur compounds, and so remove foods containing these substances and reduce inflammation.

## 5. Ethical Issues

The willingness of individuals to engage in genetic testing that would enable a personalised diet/nutrition approach will depend on how they understand the benefits this would bring compared to their perceived privacy risk [264]. Genetic testing for individuals can now be readily accessed with a number of commercial applications available (e.g., Fitgenes, 23andMe) [265,266]. The availability of genetic testing also raises a number of questions: Who should hold and have access to this information? How safely is the information kept? How informed is the consent for consumers? [266]. The protection of an individual’s genetic information and the validation of genetic tests needs to be guaranteed to ensure consumer safety. Health practitioners also need to be able to understand the role of genetic testing in their clinical practice so that they can integrate this information appropriately and offer a personalised nutrition approach.

## 6. Conclusions

This review gives examples of genotypes and genetic variants that have been associated with nutrient deficiencies, the Western diet, and food intolerances. Variation in genotypes could help explain the variability in response to food that is associated with IBD. We need to identify the variants associated with adverse reactions to food, as is currently being done in pharmacogenetic research with respect to drugs. Identification of gene variants would allow more precise tailoring of diets to avoid exacerbating malnutrition in vulnerable IBD groups. Use of the personalised nutrition approach can also more accurately identify individual food intolerances, help ameliorate abdominal symptoms and improve the quality of life for people with IBD. Further nutrigenomic research can continue to inform and improve the personalised nutrition approach in IBD. In the future, this information could be combined with data on individuals such as their gut microbiome, dietary patterns, exercise routines, blood parameters and anthropometric measurements. This combination of information could be used to build machine-learning algorithms to predict which foods/meals reduce inflammation and abdominal symptoms. This may also help to establish the dietary patterns and lifestyles that are most protective against IBD in those genetically susceptible to IBD. This combination of data may also help individuals with established IBD to tailor their diet and lifestyle to avoid exacerbating their signs and symptoms, and promote longer periods of remission. The protection of an individual’s genetic information and the validation of genetic tests needs to be guaranteed to ensure consumers’ safety when using this information with a personalised nutrition approach.

## Figures and Tables

**Table 1 nutrients-11-01532-t001:** Nutrient deficiencies in adults and adolescents with inflammatory bowel disease (IBD).

Reference	Vagianos et al. [9]	Stein and Bott [10]	Hartman et al. [15]
	Outpatient	Inpatient	Outpatient
**Year of Investigation**	2007	2008	2016
**Number of Participants**	71	na	68
**Gender (Male, Female)**	32, 52	na	na
**Age (years)**	37.6 ± 14.3	Adult	13.9 ± 3.2
**Country**	Canada	Germany	Israel
**Dietary Measurement**	FFQ, 4-day records, Anthropometric	na	3-day records, Anthropometric
**Laboratory Measurement**	Biomarkers for nutrients (CD)	na	Biomarkers for nutrients
**Dietary Deficiency**			
**Measures**	**% <66% RFI**	**% Inadequate Intake**	**% <80% of RDA**
**Vitamin B6**	4.2	na	10
**Vitamin B12**	5.6	48	12
**Folate**	20	56–62	34
**Vitamin C**	9.9	na	31
**Vitamin A**	30	11–50	57
**Vitamin D**	38	23–75	25
**Vitamin E**	59	na	65
**Calcium**	25	13	79
**Iron**	14	25–50	22
**Magnesium**	na	14–33	69
**Zinc**	8.5	40	28

CD: Crohn’s disease; FFQ: food frequency questionnaire; na: not available; RFI: dietary reference intake; RDA: required daily allowance.

**Table 2 nutrients-11-01532-t002:** Nutrients associated with healthy mucous membranes [20].

Water Soluble Vitamins	Fat-Soluble Vitamins	Minerals
Vitamin B_2_ (riboflavin)	A	Calcium
Vitamin B_3_ (niacin)	D	Fluoride
Vitamin B_6_ family	E	Zinc
Vitamin B_12_ (cobalamin)		Iron
Folic acid (folate)		
Vitamin C (ascorbic acid)		

**Table 3 nutrients-11-01532-t003:** Western diet food factors, effects related to IBD and associated genes. LC-PUFA: long-chain polyunsaturated fatty acid; SCFA: short-chain fatty acid.

Western Diet Food Factors	Effects Related to IBD	Associated Genes
Foods low in resistant starch and soluble dietary fibre [47]	Increased risk of IBD [34,36,49,50] Increased risk of *E. coli* in the mucosa [60] Decreased production of SCFAs (important in epithelial barrier maintenance and immune system regulation) [69]	*FFAR2 * (The interaction of SCFAs and *FFAR2* greatly affects inflammatory processes) [21,70]
Increased consumption of saturated fatty acids with decreased consumption of LC-PUFAs from the increased intake of soy, safflower, corn and sunflower oils and reduced consumption of fish [83,92,93]	Inflammation and cancer risk impacted by gene variant influence on metabolic pathways [85,86,87]	*FADS1 and FADS2* *PPARA, PPARG* *XRCC1, SCD1* [94,95,96,97,98,99,100,101,102,103,104]
Increased fructose consumption [25,119,120]	Associated with the rise in obesity, diabetes, metabolic syndrome, IBD, coronary heart disease [105,106,107,108,109,110,111,112,113,114,115,116] Linked to intestinal symptoms of bloating, abdominal pain, diarrhoea and fructose intolerance in IBD [112] Possible effect on vascular inflammation from infant formulas containing fructose [118]	*TXNIP* [128] (associated with promoting inflammation in endothelial cells, mediating hepatic inflammation, and regulating NF-κB) [129,130,131,132,133]
Increased use of infant formula, artificial sweeteners, food emulsifiers and antibiotics [167,168]	Reduction in gut microbiotal diversity [44,45] Increase in antibiotic resistance [168]	Genes in the microbiome [168]

**Table 4 nutrients-11-01532-t004:** Sources of foods high in fermentable oligo-, di- and mono-saccharides and polyols (FODMAPs) [185,186,187,188,189,190,191].

FODMAPs	Examples of Food Sources	Examples of Associated Genes
Fructose	Honey, mangoes, watermelon, grapes, fruit juices, high-fructose corn syrup	*TXNIP* with fructose that is linked to intestinal symptoms of bloating, abdominal pain, diarrhoea in IBD [125,138,139,140,141]
Lactose	Dairy products (e.g., milk, ice-cream, yoghurt, soft cheeses)	Lactose intolerance associated with variants of the *LCT* gene [191,192]
Fermentable oligosaccharides	Brussels sprouts, broccoli, cabbage, peas, beetroot, garlic, leeks, onions, wheat or rye bread, pasta, couscous, chickpeas, lentils	*GSTMl* and *GSTTl*, and *DIO1* variants, with tolerance of brassica vegetables [193,194,195,196]; *HLA-DQA1* and *HLA-DQB1* variants with gluten intolerance [197]
Polyols	Stone fruits, apples, pears, prunes, avocados, mushrooms; sweeteners: mannitol, sorbitol, xylitol, erythritol and isomalt.	*OCTN1* with mushrooms [198]

**Table 5 nutrients-11-01532-t005:** Differences in glucosinolate structures.

Glucobrassicin (in bok choy)	Glucoraphanin (in broccoli)
Indole-3-	Sulforaphane
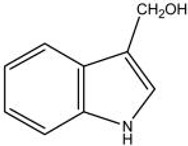	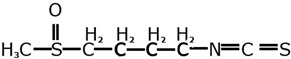

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
