# Peer review of "A Personalised Dietary Approach—A Way Forward to Manage Nutrient Deficiency, Effects of the Western Diet, and Food Intolerances in Inflammatory Bowel Disease"

_nutrients, 2019, doi:10.3390/nu11071532_

Round 1
Reviewer 1 Report
EXTENSIVE EDITING OF ENGLISH AND STYLE IS REQUIRED.
TITLE NEEDS TO BE BETTER STATED.
SOME TYPOS ARE IN THE TABLES

Author Response
1. Extensive Editing of English and style ire required
Response: The manuscript has been re-ordered to improve the style and editing done
2. Title needs to be better stated
Response: The title has been changed to better reflect the contents of the manuscript
3. Some typos in the table
Response: Typos in the table have been corrected.
Reviewer 2 Report
This paper is a review of food intolerances associated with IBD, and includes a discussion of potential gene-diet interactions that could affect nutrient response. This review requires significant revision to better achieve its objective.
Firstly, FODMAPS for IBD have been discussed extensively in recent review articles, and the extensive discussion in the present article is not necessary, unless it specifically links genotypes to FODMAPS. Presently, no examples of such a link are provided. Likewise for other potential food intolerances associated with IBD, a clear link to genotype is not provided. The article should be reduced significantly to focus on genotype-diet interactions with IBD to achieve the stated objective.
Furthermore, throughout the paper, risk and relapse in IBD are discussed closely. For the aims of this review, I would suggest reducing the focus on “risk” or just breaking “risk” and “relapse” into separate sections. In the present form, this approach is confusing. Also, throughout the discussion, I would suggest emphasizing the measures and scoring of symptoms and biomarkers of disease risk to illustrate the magnitude of differences genotype makes on these symptoms. I would suggest including an additional link between polymorphisms in drug metabolizing enzymes and xenobiotic bioavailability in the colon (curcumin/anthocyanins).
Other specific concerns are:
Section 2: In general, this section seems to overstate the certainty of the FODMAP approach for IBD.
LN 44: “…(CD) risks.” Please provide citation at first mention.
LN 74: “fermentable oligos…” FODMAP is more than this, as described in the introduction.
LN 99: It may be worthwhile to note which polyols are fermented by gut microbiota, and which are absorbed in the gut.
LN 100: The heading in the first column should be corrected, since “high foods” are not listed in this column. Also, apples and pears are more common foods that have naturally occurring polyols.
LN 111: I think that “toxic” could be a loaded word here… Please describe the specific physiological changes that could be considered detrimental.
LN 111: “This alters…” It is not clear if this should be a citation or a hypothesis put forward by the authors.
LN 112: “Commonly occurs…” Please provide a primary literature citation for this.
LN 121: “…a number of trials.” I think here, it would be best to cite the primary literature and the corresponding meta-analyses. Haven’t there been only two trials of FODMAPS for IBD?
LN 124: These citations are not the primary literature, rather reviews.
LN 182: The citations from Table 2 appear to be missing from the reference list. Please provide the number of participants from Stein 2008 reference, as well as all abbreviations in a footnote. The headings/units should be reorganized for clarity. E.g. it is not clear what the numbers are referencing as is, and seem to be different between studies.
Table 3: Please provide citations for this table.
LN 203: The first paragraph here requires clarification. My understanding is that IBD is most frequently diagnosed in young adults, so how is the aging population relevant here? Also how would receiving better healthcare increase IBD risk (LN 212)?
LN 234-5: This sentence should be reworded for clarity.
LN 249: Please mention the model organism here for clarity.
LN 253: In contrast to diet-gene interactions discussed subsequently in the paper, FFAR2 appears to link fiber to inflammation, not specifically change the risk of IBD food intolerances. The specific link from FFAR2 to diet and inflammation should be described.
LN 277: I think that it is reasonable to propose this link, but what polymorphisms would be expected to modify risk here? Isn’t the hypothesis of this paper is that genes affect the risk of food intolerances in IBD?
LN 283: Sec. 8.2 does not propose how specifically SFA consumption is linked to food intolerances or IBD relapse.
LN 299: “L-c” here is a different abbreviation than above.
LN 311: Please provide the human equivalent doses here.
LN 305: Sec 4.3 also does not link IBD relapse/intolerances specifically to genotype. This section appears to be extraneous to the goals of the review.
LN 344: Please define genes abbreviations.
Sec 4.4 does not specifically link IBD relapse/intolerance to epigenetic changes. This section is extraneous unless it can satisfy the objective of the review.
LN 378: Also milk proteins can stimulate intestinal barrier function.
LN 401: I think this point warrants further elaboration. Do intolerances precede IBD or develop at the onset? Are the authors arguing that precise genetic analysis can ultimately inform dietary recommendations for those with IBD?
LN 422-4: Please reword this for clarity.
Sec 5.2: I think a brief summary of this section contrasting the nutritional needs and lactose intolerance would be justified. Would lactose-free dairy products provide benefits? There are studies on the benefits of dairy yogurt or probiotics for IBD.
LN 465-6: Please provide a reference here.
LN 458: Again this section does not specifically link different genotypes to IBD relapse or intolerances.
LN 514-5: It is not clear why fructose malabsorption would increase IBD risk here. Wouldn’t it need to be absorbed to stimulate TXNIP?
LN 613-4: I would recommend deleting the reference to DSS here, since it would not be consumed in the diet. If that was the case, why not mention methionine or isothiocyanates again?
LN 653: This section seems out of place, considering the emphasis on foods. It may be more effective to mention the drug-genotype interactions initially in the review to set the stage for diet-genotype interactions.
LN 691-2: This seems overstated based on the information presently available.
CHECK FUT2 [30-32]
LN 359 [133,124] Do emulsifiers reduce microbial diversity?
CHECK [160] Cheese + fat content and IBD
CHECk [191] I Fructose link to thioredoxin
CHECK [201]
Author Response
1. This paper is a review of food intolerances associated with IBD, and includes a discussion of potential gene-diet interactions that could affect nutrient response. This review requires significant revision to better achieve its objective.
Response: The manuscript has been significantly revised and has been re-ordered to achieve its objectives
2. Firstly, FODMAPS for IBD have been discussed extensively in recent review articles, and the extensive discussion in the present article is not necessary, unless it specifically links genotypes to FODMAPS. Presently, no examples of such a link are provided. Likewise for other potential food intolerances associated with IBD, a clear link to genotype is not provided. The article should be reduced significantly to focus on genotype-diet interactions with IBD to achieve the stated objective.
Response: Sections have been removed, the FODMAPS section has been placed later in the manuscript and also shortened, and evidence has been placed to support nutrient-gene interactions.
3. Furthermore, throughout the paper, risk and relapse in IBD are discussed closely. For the aims of this review, I would suggest reducing the focus on “risk” or just breaking “risk” and “relapse” into separate sections. In the present form, this approach is confusing. Also, throughout the discussion, I would suggest emphasizing the measures and scoring of symptoms and biomarkers of disease risk to illustrate the magnitude of differences genotype makes on these symptoms. I would suggest including an additional link between polymorphisms in drug metabolizing enzymes and xenobiotic bioavailability in the colon (curcumin/anthocyanins).
Response: (a) The use of the term disease risk and relapse have been used more accurately in the manuscript. (b) The suggestion re measures and scoring- this would require an extra section and much more detailed explanation (c) This is a topic in its own right and has been reported in the paper by Briguglio, Matteo et al.; Food Bioactive Compounds and Their Interference in Drug Pharmacokinetic/Pharmacodynamic Profiles Pharmaceutics, 2018, 10, 4, 277, .
4. Other specific concerns (responded to individually)
Section 2: In general, this section seems to overstate the certainty of the FODMAP approach for IBD.
LN 44: “…(CD) risks.” Please provide citation at first mention. Inserted line 67
LN 74: “fermentable oligos…” FODMAP is more than this, as described in the introduction.
This section has been revised (now in section 3) and all categories of FODMAPS given
LN 99: It may be worthwhile to note which polyols are fermented by gut microbiota, and which are absorbed in the gut.
This section has been shortened as previously advised so not included
LN 100: The heading in the first column should be corrected, since “high foods” are not listed in this column. Also, apples and pears are more common foods that have naturally occurring polyols.
The heading has been revised as are the examples of food in each section
LN 111: I think that “toxic” could be a loaded word here… Please describe the specific physiological changes that could be considered detrimental.
The word toxic has been replaced and an example relating to physiological changes with diarrhoea has been included- line 432
LN 111: “This alters…” It is not clear if this should be a citation or a hypothesis put forward by the authors.
The citation has been included -line 432
LN 112: “Commonly occurs…” Please provide a primary literature citation for this.
Primary literature citations have been included- line 434
LN 121: “…a number of trials.” I think here, it would be best to cite the primary literature and the corresponding meta-analyses. Haven’t there been only two trials of FODMAPS for IBD?
This sentence has been corrected -line 442
LN 124: These citations are not the primary literature, rather reviews.
A primary source has been placed in here- line 197
LN 182: The citations from Table 2 appear to be missing from the reference list. Please provide the number of participants from Stein 2008 reference, as well as all abbreviations in a footnote. The headings/units should be reorganized for clarity. E.g. it is not clear what the numbers are referencing as is, and seem to be different between studies.
Vagianos et al. [9], Stein and Bott [10] Hartman et al. [15] are in the reference list. No numbers were available from the report on the Stein study. Abbreviations are now in the legend under the table. The table has been re-organised for clarity.
Table 3: Please provide citations for this table.
Citations have been included - line 133
LN 203: The first paragraph here requires clarification. My understanding is that IBD is most frequently diagnosed in young adults, so how is the aging population relevant here? Also how would receiving better healthcare increase IBD risk (LN 212)?
I have removed the phrase re aging population and included a clarification re better health care- line 177
LN 234-5: This sentence should be reworded for clarity.
The sentence has been removed
LN 249: Please mention the model organism here for clarity.
The sentence includes the subject of this study - Crohn’s disease patients - line219
LN 253: In contrast to diet-gene interactions discussed subsequently in the paper, FFAR2 appears to link fiber to inflammation, not specifically change the risk of IBD food intolerances. The specific link from FFAR2 to diet and inflammation should be described.
This link has been added with the inclusion of a new Table 4 (the original Table4 has been removed) and new reference added - line 237
LN 277: I think that it is reasonable to propose this link, but what polymorphisms would be expected to modify risk here? Isn’t the hypothesis of this paper is that genes affect the risk of food intolerances in IBD?
The research on this genes has yet to be completed to identify possible gene variants that could be expected to modify the risk. The hypothesis of this paper is that genes, genotypes and or gene variants affect can influence food intolerances in IBD.
LN 283: Sec. 8.2 does not propose how specifically SFA consumption is linked to food intolerances or IBD relapse.
The manuscript has been reorganised. This section is linked to the effects of the Western diet and how the reduction in LC –PUFA and increase in SFA (saturated fatty acids) influences genes through gene variants. Metabolic pathways are influenced depending on a person’s gene variants, and this can increase or decrease inflammation and cancer risk.
LN 299: “L-c” here is a different abbreviation than above.
This has been corrected
LN 311: Please provide the human equivalent doses here.
The USA Federal Drug Administration – FDA- acceptable daily intake for humans is 5 mg/kg/day) is in the text
LN 305: Sec 4.3 also does not link IBD relapse/intolerances specifically to genotype. This section appears to be extraneous to the goals of the review.
The goals of the review have been clarified and this section 4.3 in the context of the Western diet (part of Table 4) has more congruence with these goals
LN 344: Please define genes abbreviations.
Abbreviations have been explained
Sec 4.4 does not specifically link IBD relapse/intolerance to epigenetic changes. This section is extraneous unless it can satisfy the objective of the review.
This section has been removed
LN 378: Also milk proteins can stimulate intestinal barrier function.
A sentence about this has now been included lines 355-6
LN 401: I think this point warrants further elaboration. Do intolerances precede IBD or develop at the onset? Are the authors arguing that precise genetic analysis can ultimately inform dietary recommendations for those with IBD?
(a) Do intolerances precede IBD or develop at the onset?
This question has yet to be elucidated – analysis of longitudinal studies may be able to answer this. The interplay of multiple factors play a role in the pathogenesis of IBD and individual’s gene variants are part of the story.
(b) Are the authors arguing that precise genetic analysis can ultimately inform dietary recommendations for those with IBD?
Further nutrigenomic research can continue to inform and improve the personalised nutrition approach (by identifying genotypes and gene variants associated with nutrients) for individuals in IBD. The outcomes of this research (as stated in the conclusion) can be combined with data on individuals such as their gut microbiome, dietary patterns, exercise routines, blood parameters and anthropometrics. This combination of information could be used to build machine learning algorithms to predict which food/meals reduce inflammation and abdominal symptoms. This may also help establish the dietary patterns and lifestyles that are most protective against IBD in those individuals who are genetically susceptible to IBD.
LN 422-4: Please reword this for clarity.
This has been reworded – lines 493-495
Sec 5.2: I think a brief summary of this section contrasting the nutritional needs and lactose intolerance would be justified. Would lactose-free dairy products provide benefits? There are studies on the benefits of dairy yogurt or probiotics for IBD
There is a market for lactose free dairy foods but there are also a number of non-dairy foods that also contain calcium, e.g. fortified soy, almond and rice milks, tofu, sardines, almonds, sesame seeds, broccoli, and fortified breakfast cereals and juices and this has been included in the manuscript.
LN 465-6: Please provide a reference here.
The references are now in place – lines 290, 291 & 294
LN 458: Again this section does not specifically link different genotypes to IBD relapse or intolerances.
The link is made with the discussion of the influence of the TXNIP gene and in Table 4 and Table 5
LN 514-5: It is not clear why fructose malabsorption would increase IBD risk here. Wouldn’t it need to be absorbed to stimulate TXNIP?
This section is about the effect of fructose and sugar consumption –and when too much fructose is absorbed this stimulates the TXNIP and this is associated with promoting inflammation and regulating NF-κB the pro-inflammatory signaling pathway. This suggests when the gut is inflamed as with IBD fructose is not as well tolerated .
LN 613-4: I would recommend deleting the reference to DSS here, since it would not be consumed in the diet. If that was the case, why not mention methionine or isothiocyanates again?
This sentence has been deleted
LN 653: This section seems out of place, considering the emphasis on foods. It may be more effective to mention the drug-genotype interactions initially in the review to set the stage for diet-genotype interactions.
This has been done
LN 691-2: This seems overstated based on the information presently available. L
This has been reworded – line 652
CHECK FUT2 [30-32]
These have been checked in the literature
LN 359 [133,124] Do emulsifiers reduce microbial diversity?
Yes and this has been added in the text
CHECK [160] Cheese + fat content and IBD
Yes checked with literature
CHECk [191] I Fructose link to thioredoxin
Yes checked
CHECK [201]
Yes checked
Reviewer 3 Report
Dear Editor,
thank you for the opportunity to revise this very interesting review article.
In this manuscript, the authors aimed at discussing the connection between different food intolerances/avoidance and the appearance of inflammatory bowel disease (IBD). The topic is interesting and the scientific approach used to draw up this manuscript is very comprehensive and complete, as it carefully reviews a variety of dietary habits in association to IBD, and take into consideration both clinical and molecular aspects.
I recommend the authors to carefully check the manuscript for language editing, as some typographical and grammar errors are present in the whole text.
I would also ask the authors if it could be possible to provide a table/figure (one or more) summarizing the interconnection among food/nutrients, genetic susceptibility and the onset of IBD. I think this could help the reader to better comprehend the number of information provided in this review article.
Author Response
Dear Editor,
Thank you for the opportunity to revise this very interesting review article.
In this manuscript, the authors aimed at discussing the connection between different food intolerances/avoidance and the appearance of inflammatory bowel disease (IBD). The topic is interesting and the scientific approach used to draw up this manuscript is very comprehensive and complete, as it carefully reviews a variety of dietary habits in association to IBD, and take into consideration both clinical and molecular aspects.
I recommend the authors to carefully check the manuscript for language editing, as some typographical and grammar errors are present in the whole text.
Response: This has been done
I would also ask the authors if it could be possible to provide a table/figure (one or more) summarizing the interconnection among food/nutrients, genetic susceptibility and the onset of IBD. I think this could help the reader to better comprehend the number of information provided in this review article.
A table has been included (Table Four) replacing the previous table to clarify these points
Round 2
Reviewer 1 Report
In this review article, it would be nice to include a paragraph to discuss the ethics and confidentiality issues of genotype and genetic variants of a patient or health individual in a so called personalized diet/nutrition approach.
In the attachment, please see some recommendations and terms to be clarified.

Author Response
Thank you for your constructive feedback and the time your have put in to improve this manuscript. The responses to your comments are below.
Comment One:
In this review article, it would be nice to include a paragraph to discuss the ethics and confidentiality issues of genotype and genetic variants of a patient or healh individual in a so called personalized diet/nutrition.
Response:
A paragraph has been included at the end of the discussion, before the conclusion.
Comment Two
In the attachement, please see some recommendations and terms to be clarified.
Response:
The responses to the comments in this pdf are attached. Where no comments have been made but areas highlighted - modifications have been made to enhance the readability of the manuscript and increase clarity.

Reviewer 2 Report
The paper has been extensively modified. Just two outstanding comments here:
LN 647-650: These conclusions were not established by the review here, I would suggest removing them.
The Table 6 images should be provided at higher resolution and I believe "carbinol" has been omitted.
Author Response
Thank you for your review and responses to improve the manuscript. Responses to your comments are below.
Comment One:
LN 647-650 These conlusions were not established by the review here. I would suggest removing them.
Response:
These lines have been removed.
Conmment Two:
The Table 6 images should be provided at higher resolution.
These images have been replaced with images of higher resolution.